# Conformational and Structural Changes in Chickpea Proteins Caused by Simulated Salivary Alterations in the Elderly

**DOI:** 10.3390/foods12193668

**Published:** 2023-10-05

**Authors:** Ingrid Contardo, Fanny Guzmán, Javier Enrione

**Affiliations:** 1Biopolymer Research and Engineering Laboratory (BiopREL), School of Nutrition and Dietetics, Faculty of Medicine, Universidad de los Andes, Chile, Monseñor Álvaro del Portillo 12.455, Las Condes, Santiago 7620086, Chile; jenrione@uandes.cl; 2Centro de Investigación e Innovación Biomédica (CIIB), Universidad de los Andes, Chile, Monseñor Álvaro del Portillo 12.455, Las Condes, Santiago 7620086, Chile; 3Núcleo Biotecnología Curauma, Pontificia Universidad Católica de Valparaíso, Valparaíso 2373223, Chile; fanny.guzman@pucv.cl

**Keywords:** elderly salivary alterations, oral electrolytes, chickpea proteins, whey proteins, secondary structure, circular dichroism

## Abstract

The impact of salivary alterations on chickpea protein structure in the elderly has not been well documented. This study aimed to understand the role of simulated salivary alterations in the conformational properties and secondary structure of the chickpea protein isolate (CPI). Whey protein isolate (WPI) was used as the reference. Protein dispersions (10%) were subjected to in vitro oral processing under simulated salivary conditions in both the elderly and adult subjects. Proteins and their oral counterparts were characterized in terms of their composition, charge, size, solubility, water absorption, molecular weight (MW), and secondary structure (Circular Dichroism and Raman spectroscopy). Under condition of simulated oral digestion in the elderly population, the ordered secondary protein structure was significantly affected, decreasing α-helix by ~36% and ~29% in CPI and WPI compared to the control (adult) population, respectively. An increase in the unordered random coil state was observed. These results could be attributed to an increase in electrolytes in the salivary composition. The structure of CPI is more stable than that of WPI because of its higher MW, more rigid structure, less charged surface, and different amino acid compositions. This study is meaningful in understanding how alterations in the elderly oral system affect protein conformation and is expected to improve the understanding of plant-based protein digestibility.

## 1. Introduction

Legume proteins differ from their animal counterparts in terms of nutritional quality; they are limited in sulfur amino acids (e.g., lysine, methionine, cysteine, and tryptophan) and have lower digestibility [1]. The conformational aspects of proteins influence their physicochemical properties, functionality, and digestibility [2]. Digestion factors such as pH, ionic strength, electrolytes, and enzyme concentration can modify the surface properties and alter the structural organization of legume proteins [3,4]. Thus, aging-related changes at the oral level (e.g., the shift in pH and ionic strength) could affect the overall legume protein structure differently and determine their subsequent gastrointestinal digestibility.

Chickpea exhibits great potential as a valuable food component owing to its high protein content (23–29%), significant amounts of essential amino acids (except sulfur-containing amino acids), adequate protein digestibility (~80%), and low cost [5,6]. Specifically, chickpea protein isolates (CPI) contain a high fraction of globulins (salt-soluble), representing 80% of the total seed proteins, and a lower fraction of albumins (water-soluble) (~20%), whereas glutelins (soluble in dilute acid/base) (~18%) and prolamins (alcohol-soluble) (<5%) are minor protein fractions. The isoelectric point (pI) varies between 4.0 to 5.0 in different cultivars [7], and CPI exhibits a net negative charge (close to −27 mV) at neutral pH, which is explained by the fact that this pH value is well above their pI. In terms of amino acid composition present in chickpea proteins, it is reported that ~29% are acidic, ~33% are hydrophobic, ~18% are polar uncharged, and ~19% are basic [8]. The destabilization of chickpea proteins depends on an adequate balance between their overall surface charge, secondary structure, molecular weight, and unfolding, which can be affected by environmental factors, such as pH, ionic strength, temperature, time, and enzyme concentration throughout the digestion process [9].

Oral processing is the first stage of the digestion process, in which different foods are broken down or mixed, and a bolus is formed. Dietary proteins are exposed to oral pH, ionic strength, electrolytes, and the action of salivary enzymes, among other parameters. These extrinsic conditions, especially electrolytes, might alter the secondary structural components of chickpea proteins, modifying their conformations; however, scarce information has been reported on structural changes at the oral level. A shift in pH may result in the reorganization of the secondary structure of globular proteins. For instance, it has been reported that ~80% of the secondary structures of CPI isolates are likely to show α-helix ordering at pI (pH ~5.0). However, at neutral pH values, the β-sheet structures and random coil composition increase at the expense of α-helix (~48%) [10]. Likewise, it has been observed that acidic pH decreases the β-sheet content and increases the random coil state, suggesting protein unfolding, since large net charges are induced, and therefore repulsive forces increase [11]. In addition, the presence of sodium chloride (NaCl) increases the β-sheet content of wheat gluten proteins, mainly at the cost of a decrease in α-helix content. This salt had no dose-dependent effect on the increase in β-sheet content, but it might induce the formation of hydrogen bonds. Secondary structural changes were mainly determined by the protein fraction [12].

Changes in saliva composition in elderly individuals (>60 years old) in comparison with adults are associated with a significant increase in the concentration of inorganic elements such as chloride (Cl^−^), potassium (K^+^), and phosphate (P), except for calcium (Ca^2+^), which decreases. Also, an increase is observed in the concentration of organic components, α-amylase, uric acid, lysozyme, and immunoglobulin (IgA), except for mucin (muc1 and muc2) and lactoferrin, which decrease [13]. In addition, elderly subjects usually show changes in salivary pH. The fluctuation in salivary pH is dependent on gender, clinical status, and the severity of periodontal disease, as well as on the flow rate of saliva following unstimulated and stimulated conditions [14]. Stimulated saliva represents secretion during food intake (physiological stimulation), and unstimulated saliva reflects the basal salivary flow rate. In healthy elderly individuals, the pH values in unstimulated saliva can change from acid to neutral (6.0 to 7.0), while in stimulated saliva, these values fluctuate between neutral to alkaline (7.0 to 8.0) [15].

Accordingly, the secondary structure, as well as the surface properties of chickpea proteins, would be affected differently by the salivary alterations associated with salivary electrolytes and pH shift in elderly individuals. These structural changes in orally treated proteins might further impact protein–protein, protein–water, or protein–enzyme interactions during subsequent gastrointestinal digestion. Therefore, this study aimed to understand the effect of simulated salivary alterations in the elderly on changes in surface properties and the secondary structure of the CPI. The proteins were subjected to in vitro oral processing, simulating both elderly and adult salivary conditions (adjusted pH, chloride concentration, and α-amylase activity). In addition, we characterized the physicochemical changes in CPI and whey protein isolate (WPI), which was used as a reference protein. This complete characterization of CPI processed at the oral level will serve as a baseline for future complementary studies aimed at identifying ways to improve plant-based protein digestibility in the general population, particularly in elderly individuals.

## 2. Materials and Methods

### 2.1. Materials and Chemicals

Dehulled chickpea grains (Sociedad Industrial y Comercial Díaz y Cía. Ltd., Santiago, Chile) were stored in sealed plastic bags at approximately 5 °C. Whey protein isolate (WPI) (BIPRO, Davisco Foods International, Le Sueur, MN, USA) was used as a milk-derived reference protein. Alpha-amylase (A3176, type VI-B, ≥5 units/mg solid) was obtained from Sigma-Aldrich. The pH range for α-amylase activity was 5.5 to 8.0, with an optimum pH of 7.0. Enzyme activity was determined as described by Brodkorb et al. (2019) [16]. The rate at which maltose is released from starch is determined by its ability to reduce 3,5-dinitrosalicylic acid. One unit releases 1.0 mg of maltose equivalent from starch in 3 min at pH 6.9 and 20 °C. The amount of α-amylase in the simulated salivary fluid was adjusted to obtain the required enzyme activity (75 or 150 U/mL for adults and the elderly, respectively) in the final simulated salivary mixture. All other chemicals used were of analytical grade.

### 2.2. Extraction of Chickpea Protein Isolates (CPI)

The dehulled grains were ground into fine flour (<500 µm) using a cross-beater mill (Stainless Steel Pulverisette 16; Fritsch, Idar-Oberstein, Germany). These were fractionated into five fractions of different particle sizes to produce protein-enriched chickpea flour. The flour was passed through an electromagnetic vibrator (Filtra S.A) equipped with five mesh screens (425, 300, 212, 150, and 106 µm). The fraction of flour enriched in proteins (212–300 µm) was defatted and used. It was defatted according to Wang et al. (2020) [17] with modifications. This fraction was then immersed in hexane (1:5 by weight) with constant agitation (150 rpm) for 3 h at 37 °C. The upper portion of the extract was discarded. This process was repeated twice. Hexane was then evaporated at room temperature for 24 h. The moisture content (% wet basis) was determined according to the AOAC (2019) [18].

Alkaline extraction. Defatted chickpea flour (~50 g) was mixed with 1000 mL of 5 mM NaOH solution, and the pH was adjusted to 9.0 using 1 N NaOH. The solution was magnetically stirred for 3 h at room temperature with constant agitation (350 rpm). Then, the slurry was centrifuged at 5000 rpm for 15 min to precipitate the insoluble components. The pH of the supernatant was adjusted to pH 4.5 (referenced pI of chickpea proteins) using 2 N HCl. This sample was centrifuged at 5000 rpm for 15 min to precipitate soluble chickpea proteins. These proteins were then mixed with distilled water (1:2 precipitated/water) and the pH was adjusted to 7.0 [19]. CPI was frozen at −80 °C and lyophilized using a vacuum freeze dryer (Labconco, Kansas City, MO, USA). The samples were hermetically kept at −20 °C for future analysis.

### 2.3. Proximate Analysis

The proximate composition of the samples was determined according to the methods described by AOAC (2019) [18] as follows: moisture content by oven drying at 105 °C for 24 h, fat content by Soxhlet, protein content by the Kjeldahl method (%N × 6.25), and ash content using a muffle furnace at 550 °C. Non-nitrogenous extract (NNE) was determined by weight difference by subtracting the total weight of the other components quantified in the samples.

### 2.4. Amino Acidic Profile by HPLC

The concentration of amino acids was determined by reversed-phase high-performance liquid chromatography (RP-HPLC) following the methodology of Enrione et al. (2020) [20] with modifications. The amino acids were separated and quantified using a chromatograph (Waters 600 controller, Milford, MA, USA) with a diode array detector (Waters 996) using an RP18 column (150 mm × 4.6 mm, particle size 5 µm) (Luna, Phenomenex, Los Angeles, CA, USA). The elution of compounds was achieved using a gradient elution method with two mobile phases. Mobile phase A consisted of an acetate buffer adjusted to pH 5.8, while mobile phase B comprised acetonitrile: water (60:40). The oven temperature was maintained at 40 °C, and the flow rate was set at 1.5 mL/min. A sample volume of 20 µL was injected for analysis. The amino acid content was reported as g/100 g of protein.

### 2.5. Surface Charge and Hydrodynamic Diameter Measurements

The zeta potential values of the samples (0.1 mg/mL) were determined as a function of pH (2.0 to 12.0, using 1 N HCl and 1 N NaOH) using an automatic pH titrator attached to a Dynamic Light Scattering (DLS) unit (Zetasizer Pro, Malvern, UK) [21]. Three independent runs with three measurements at each pH were performed (*n* = 9). Also, the hydrodynamic diameter (d.nm) of samples (1 mg/mL) was measured by DLS. Three independent runs, with ten measurements, were performed (*n* = 30).

### 2.6. Water Absorption and Solubility

#### 2.6.1. Water Absorption Index (WAI)

The WAI was determined as described by Contardo et al. (2018) [22] with some modifications. Approximately 0.2 g of the sample was hydrated under water-excess conditions using constant agitation (150 rpm) for 30 min at 37 °C, maintaining a sample-to-water ratio of 1:100. The solution was centrifuged at 5000 rpm for 30 min at 37 °C. The precipitate was weighed to determine the amount of water absorbed per gram of dry solid.

#### 2.6.2. Solubility

The solubility of three CPI obtained at different alkaline pH values (8.0, 9.0, and 10.0, samples labeled as CPI-pH8, CPI-pH9, and CPI-pH10, respectively) was evaluated as a function of pH from 2.0 to 10.0. (Appendix A). In addition, the solubility of the simulated oral-treated QPI and WPI samples was analyzed.

The supernatant (soluble solids) obtained from the WAI analysis (Section 2.6.1) was transferred to a weighed dish and then dried in a forced-air convection oven at 105 °C to a constant weight. The % of solubility was determined using Equation (1):Solubility (%) = g soluble dry solids/g dry solids(1)

In addition, the % of soluble proteins was determined by the Bradford method using a protein assay standard curve produced using BSA. Approximately 100 µL of the supernatant was mixed with 3 mL of Bradford reagent and incubated for 5 min. Absorbance was measured at 595 nm. Soluble proteins were expressed as percentages of the total protein.

### 2.7. Structural Characterization of Proteins

#### 2.7.1. Molecular Weight Distribution by Electrophoresis (SDS-PAGE)

The molecular weight (MW) distribution of the samples was determined by SDS-PAGE gel electrophoresis. All the steps for SDS-PAGE were performed as described previously by Díaz-Calderón et al. (2017) [23] using 4–20% acrylamide gradient precast gels (Bio-Rad Laboratories, Inc., Hercules, CA, USA). The samples were diluted with distilled water to a final concentration of 3 mg protein/mL. Samples (30 µL) were mixed 1:1 with sample buffer and 10 µL of 2-mercaptoethanol. All samples were heated at 100 °C for 5 min before loading. Subsequently, 10 μL of the sample solution was loaded into the electrophoresis lane. Standard molecular weight markers within the 10–250 kDa range were used (KaleidoscopeTM, Precision Plus Protein StandardsTM; Bio-Rad). The electrophoresis was run at 90 V, and the resulting gel was stained with 0.25% Coomassie blue G250.

#### 2.7.2. Changes in Protein Structure by Circular Dichroism (CD)

Solutions of protein isolates were prepared at varying concentrations (0.2 to 0.5 mg/mL in sodium phosphate buffer, pH 7.0) and then centrifuged at 10,000× *g* for 5 min, and the supernatant was filtered using a 0.45 µm syringe filter to remove suspended particles. The CD spectra were recorded in the far-UV range (190–250 nm) with a spectropolarimeter (Jasco J-815, Jasco, Easton, MD, USA) coupled to a Peltier JASCO CDF-426S/15 system for temperature control at 25 °C and 37 °C. Quartz cuvettes with 0.1 cm optical path length were used. The scanning speed, data pitch, and bandwidth were set to 50 nm/min, 0.25 s, and 1.0 nm, respectively. Three scans were averaged to obtain the spectrum. The CD data were then analyzed using Spectra Manager software version 2.0 and expressed in terms of ellipticity (*Mol. Ellip*.) as a reference for the composition after subtracting the contribution of the buffer from each spectrum [24].

#### 2.7.3. Changes in Secondary Structure by Raman Spectroscopy

Samples were analyzed using a Raman spectrometer (XploRA PLUS, Horiba Scientific, Palaiseau, France) equipped with a near-infrared laser (λ785 nm), which was set to 70 mW. All spectra were acquired using a diffraction grating with a groove density of 1200 g/mm. The laser was focused using an optical microscope (Olympus BX41, Olympus Optical Co. Ltd., Tokyo, Japan) with an X100 long-working-distance objective (PL Fluotar, NA = 0.55). Each spectrum was acquired within a wavenumber range of 200–3200 cm^−1^ using an exposure time of 30 s and three accumulations [25]. All the spectra were corrected using instrument software (LabSpec 6 software version 6.4, Horiba Scientific, Palaiseau, France) and normalized according to the protein phenylalanine peak at 1003 ± 1 cm^−1^ using Origin software (OrigingPro 2019b). Three replicates were used for each of the six positions on the surface of each sample. Protein secondary structures were determined as the percentages of α-helices, β-sheets, β-turns, and random coils.

### 2.8. Simulated Oral Processing

The in vitro oral processing involved the exposure of the protein isolate suspensions (10%, *w*/*v*) to a static method, as recommended by Brodkorb et al. (2019) [16] with modifications. The samples were mixed with simulated salivary fluid (SSF) at a 1:1 (*w*/*w*) ratio and α-amylase for 2 min. The processing was stopped by snap freezing the samples with liquid nitrogen and freeze-drying. The oral conditions and simulated salivary compositions are presented in Appendix A. The experimental conditions were based on the available physiological data of a fed state for a typical meal, which was presented and justified in detail by Brodkorb et al. (2019) [16].

### 2.9. Statistical Analysis

The results are expressed as mean ± standard deviation of triplicate samples. Experimental data were tested using the Shapiro–Wilk test to assess normality. Analysis of variance (ANOVA) was used for normally distributed datasets at a level of significance of *p* < 0.05, and non-normally distributed datasets were analyzed using the Kruskal–Wallis test. The differences between sample sets were resolved using the LSD method at 95% confidence and Statgraphics Centurion XV software (Manugistics, Inc., Rockville, MD, USA) for Windows.

## 3. Results and Discussion

### 3.1. Solubility of Proteins Affected by pH of Simulated Salivary Fluid

Stimulated saliva represents the secretion during food intake (physiologic stimulation), and unstimulated saliva reflects the basal salivary flow rate. Some studies have demonstrated that stimulated saliva becomes more alkaline with increasing age, with a maximum pH of ~8.0 [15]. Meanwhile, unstimulated saliva becomes more acidic, with an average value close to pH 6.0 in elderly individuals (>65 years) [26]. Therefore, the solubility and protein solubility of CPI and WPI subjected to varying pH conditions (associated with alterations in the elderly and adults) at 37 °C were determined. These results are shown in Figure 1, indicating that alterations in pH of simulated salivary fluid affected protein solubility and soluble solids of CPI and WPI. Specifically, protein solubility (Figure 1B) of CPI exhibited a significant decrease (*p* < 0.05) at pH 6.0 (corresponding to 46%), while at neutral or alkaline salivary pH, these values shifted towards ~100%. This pH dependency was similar for the soluble solids of the CPI evaluated (Figure 1A). In contrast, the solubility of WPI was not significantly affected by a decrease or increase in pH of simulated salivary fluid, exhibiting high solubility values of >95% at all evaluated pH values. These differences demonstrate that alterations in the salivary pH of elderly individuals might impact chickpea proteins differently than whey proteins. These can be explained by the relative contributions of ionized amino acids on the surface of proteins and the balance of hydrophobic and electrostatic interactions [27].

Poor protein solubility of CPI at pH 6.0, ranging between 35 and 50%, has been reported by other authors [8]. In this study, the low surface charge of CPI at pH 6.0 (see Figure 2A) could also negatively influence the solubility of proteins, suggesting that this pH promotes less exposure of hydrophilic zones on the surface and reinforces the importance of the protonation/deprotonation of arginine (Arg) in the solubilization of chickpea proteins. In addition, legume proteins have high hydrophobicity, which impairs solubility. The decrease in solubility at pH 6.0 might be attributed to increased exposure of the aromatic and aliphatic amino acid residues on the surface of chickpea proteins [3]. On the other hand, the higher solubility of WPI samples at pH 6.0 could be explained by the fact that the major components of whey proteins, such as β-lactoglobulin (16.7 kDa), α-lactalbumin (13.5 kDa), and bovine serum albumin (BSA) (66.5 kDa), exhibit subunits with low molecular weight, suggesting a more exposed surface [28]. Thus, charged groups could be maintained at the surface of whey proteins, promoting protein–water interactions, whereas hydrophobic interactions should be weakened, resulting in higher solubility in WPI samples [29].

### 3.2. Surface Charge of Proteins

The surface charge of proteins is defined by the ionization of the surface groups [30]. Zeta potential is a measure of the surface net electrical charge of proteins, which is relevant for their solubility, aggregation, and stability under different environments. Figure 2 shows the zeta potential values as a function of the pH of the raw CPI and WPI (Figure 2A) and oral-treated CPI and WPI (Figure 2B) after in vitro oral processing under simulated adults (-A) and elderly (-EL) salivary conditions. As shown in Figure 2A, both protein types had similar profiles with a negative surface net charge above the pI (pI = 4.5, CPI and pI = 4.4 WPI), which is consistent with the deprotonation of the exposed carboxyl groups and amino groups of the globular proteins. Similar profiles have been reported previously [3,31]. The pI of chickpea proteins can be influenced by the proportion of globulin/albumin after extraction (pI ~4.5, globulins and ~6.0 for albumins). Isoelectric precipitation mainly separates globulins [32]. Thus, a pI of 4.5, as shown by the CPI obtained in this study, suggests a greater proportion of the globulin fraction.

The CPI presented less negative surface charge values (from 23.6 to −30.6 mV) than whey proteins (WPI, from 30.0 to −31.8 mV) in almost the entire pH range studied, except at extreme pH. Accordingly, the CPI produced by isoelectric precipitation may have resulted in samples with lower deprotonation at alkaline pH up to pH 10 and reduced protonation close to and above the pI than the commercial WPI. These results can be explained by the balance of ionic hydrophilic groups and nonpolar hydrophobic amino acid residues that affect the surface charge of chickpea proteins [3,33], suggesting major exposure of nonpolar hydrophobic residues (e.g., aromatic or aliphatic) in a pH range greater than 4 or 10. Zhang et al. (2009) [10] demonstrated that environmental pH significantly affects the secondary structure of chickpea proteins, with more protein–protein interactions being promoted at a higher acid pH.

In a complementary manner, the decrease in the surface charge of CPI might be influenced by other components (e.g., lipids) (see Appendix A). It is hypothesized that the lipid content in CPI samples could alter the exposure of charged groups of proteins via a physical barrier on the surface. In addition, lipids exhibit relatively low net charges [34]. Commercial WPI resulted in proteins with slightly more negative charges, which could be attributed to their lower molecular weights. This might enhance electrostatic repulsion, promote protein unfolding, and increase protein—water interactions [35]. Moreover, Figure 2B shows the zeta potential of orally treated CPI (CPI-A and CPI-EL) subjected to a pH shift from alkaline to acidic pH range, simulating when the bolus passes from the oral to the gastric environment. These results indicated that orally treated chickpea and whey proteins exhibited negative charges when exposed to a decrease in pH from 8.0 to 6.0. Below this point, the zeta potential begins to decrease, passing from negative to positive values at a pH close to 4.0. However, the simulated saliva of the elderly had a different effect on the surface charge of chickpea proteins than that of adults. CPI-EL showed a lower pI value, at which the zeta potential changed to positive values (pI = 4.0, CPI-A and 3.5 for CPI-EL) (CPI > CPI-A > CPI-EL). The zeta potential differences might be attributed to a higher electrolyte concentration in terms of chloride content in the simulated salivary fluid of the elderly. A higher amount of negatively charged electrolytes contributed by chloride may promote electrostatic attraction between oppositely charged proteins, thereby altering the profile and charge distribution on the surface of chickpea proteins via salt ions [36]. These shifts in the surface charge of orally treated chickpea proteins could further influence their digestibility during gastric digestion.

### 3.3. Water Absorption Index (WAI)

Appendix A shows that the WAI values of CPI were significantly (*p* < 0.05) affected by variations in pH of simulated salivary fluid, considering the pH variations between the elderly and adult conditions. Thus, when the chickpea proteins were exposed to pH 6, the WAI was higher (WAI = 2.2 g/g) than that exhibited at neutral (WAI = 0.9 g/g) or alkaline (WAI = 0.6 g/g) pH. These results might be explained by the lower availability of acidic, basic, and hydrophilic groups of polar amino acids, such as Glu, Lys, or Tyr in CPI (see Appendix A), which are the primary sites for water interaction of proteins at ~37 °C. In addition, a negative correlation with solubility was observed (Figure 1B). As the system was alkalized, the WAI of chickpea proteins decreased and protein solubility increased, whereas the WAI values of WPI were near zero and were not affected by changes in pH of simulated salivary fluid in the range studied. This result could be related to the high solubility of the WPI samples.

### 3.4. Particle Size Distribution

The particle size distribution of the studied proteins was sensitive to oral changes in the ionic strength. Figure 3 shows that the particle size distribution of both CPI and WPI samples was influenced by the simulated salivary conditions of the elderly. The size distribution shifted towards larger sizes (~100 d, nm) in both protein sources: CPI-EL (Figure 3A), and WPI-EL (Figure 3B). It is possible that swelling and aggregation occur simultaneously under these conditions. Nonetheless, the CPI-EL samples exhibited lower volume (%) values for large protein particle fractions (100 d, nm) than their WPI-EL counterparts. In addition, the shape of the curve showed a narrower profile, which suggests that less aggregation could have occurred in chickpea proteins, despite being subjected to the high ionic strength generated by the presence of more monovalent ions (Cl^−^) in the simulated salivary fluid of the elderly (see Appendix A); while the simulated oral conditions of adults did not significantly (*p* > 0.05) impact CPI-A, they did affect whey proteins, promoting larger particles in WPI-A samples after oral processing. As a result, chickpea proteins appear to be more stable and prevent more interactions between protein molecules than whey proteins when exposed to high ionic strength orally. This could be explained by the higher molecular size or more rigid structure of chickpea proteins than that of whey proteins or a less charged surface (see Figure 2), resulting in less aggregation when exposed to neutral pH and low ionic strength, as found in the simulated salivary conditions of adults. In addition, different ionic strengths at the oral level had different effects on the charge distribution on the surfaces of the studied proteins.

### 3.5. Molecular Weight Distribution

SDS-PAGE showed protein profiles of CPI with a molecular weight (MW) distribution of 11 to 100 kDa, with the subunit structures shown in Figure 4. CPI exhibited characteristic electrophoretic bands for chickpea proteins [37]. This displayed an electrophoretic pattern with typical polypeptide chains corresponding to legumin and vicilin reported by other authors, such as 11S legumin (54–60 kDa), the acidic α-subunit of legumin (36 kDa), the basic β-legumin subunit (26 kDa), albumin (63 and 100 kDa), and 7S vicilin (60–65 kDa) [37,38]. There were differences in the bands between raw CPI and treated CPI under adult (CPI-A) and elderly (CPI-EL) oral conditions, which demonstrates that alterations in salivary elderly individuals can alter the MW distribution of the chickpea protein subunits. The intensities of bands close to 26 and 36 kDa (see CPI, line 2) were reduced, shifting to subunits with lower MW values in the CPI-A and CPI-EL samples, respectively. The appearance of bands with subunits of low MW exhibited by CPI-A (<15 kDa) and CPI-EL (<10 kDa) corroborates the idea that a lower molecular weight might be related to a slightly more negative charge in zeta potential analysis. A similar behavior was observed in WPI, but in this case, the decrease in the intensity of bands and shift of MW values were higher than those exhibited by the CPI counterparts. Consequently, simulated salivary conditions of the elderly impacted the MW distribution of chickpea proteins, but a more significant effect was observed on whey proteins.

### 3.6. Molecular Configuration and Secondary Structure

#### 3.6.1. Circular Dichroism

The CD spectra of CPI and WPI are shown in Figure 5. In vitro oral processing under elderly conditions altered the protein configuration of both sources. This can be observed by the increase in the unordered or random coil state in the treated samples (CPI < CPI-A < CPI-EL and WPI < WPI-A < WPI-EL) and by the spectra measured at 25 °C (Figure 5A), and 37 °C (Figure 5B). The structural properties of chickpea proteins were more stable than those of whey proteins after in vitro oral processing. The CD spectra of CPI showed a decrease in the area and a shift towards low ellipticity values (according to the arrow in Figure 5A), suggesting an increase in the unordered state of these proteins. A typical CD spectrum showing the tendency of the α-helix secondary structure involves a double minimum at 222 nm and 208 nm, and a strong maximum close to 191 nm. In the present study, CPI (red line) exhibited an α-helical secondary structure, with a maximum at 195 nm (8 × 10^6^, Mil. Ellip.) and two minima at 210 nm and 220 nm, respectively. CPI-A (blue line) exhibited a maximum at 195 nm, but decreased (4 × 10^6^, Mil. Ellip.), and the two minimums tended to disappear, thus, the structural tendency of the α-helix was lost. Finally, CPI-EL (green line) exhibited a maximum at 195 nm, which decreased even further (2 × 10^6^, Mil. Ellip.). The behavior of the minimum is very similar to that of the CPI-A samples. These changes were promoted when the temperature increased to 37 °C, indicating that physiological temperature at the oral level, alkaline salivary pH, and high electrolyte content, such as chloride and potassium, in elderly salivary conditions might favor the unfolding of chickpea proteins.

Interestingly, this destabilization was higher in the WPI. The orally treated WPI lost this characteristic of shape helical state, such as α-helix, shifting towards low minimum wavelength values after in vitro oral processing and decreasing the area of the CD curves. In WPI-A, the maximum shifted to 190 nm, and a single minimum was observed at 208 nm, which may indicate a modification of the secondary structure to a β-strand. The WPI-EL showed that both the maximum and minimum disappeared, indicating a loss of the structural trend. Thus, the spectra of WPI-A and WPI-EL were more representative of an unordered state in globular proteins. These differences between CPI and WPI could be associated with differences in amino acid composition (see Appendix A), the lower molecular weight of whey proteins, and the distribution of the ion density of oral electrolytes on the α-helix and around charged amino acid residues, thus affecting the conformational stability of both proteins differently.

Nishanthi et al. (2017) [39] discussed that at high pH values and due to high ionic strength, the ion density is localized around the α-helix and surrounds positively charged Lys residues, demonstrating that elevated salt concentration is responsible for structural differences in whey proteins. In an oral saline environment, the structural stability of proteins appears to be driven by hydrogen bonding and electrostatic interactions. The secondary structure of globular proteins is stabilized by hydrogen bonds, and the results of CD analysis suggest that oral processing in an elderly environment with alkaline salivary pH and a higher concentration of chloride (Cl^−^) alters the ordered state of the secondary structure of chickpea and whey proteins, probably due to the disruption of hydrogen bonds. This was also observed and confirmed by Raman spectroscopy.

#### 3.6.2. Raman Spectroscopy

The secondary structures of CPI and WPI proteins were analyzed using Raman spectroscopy, followed by deconvolution of the amide I band. Table 1 shows the percentage of the secondary structure of CPI and WPI proteins before and after in vitro oral processing. The percentage values of the secondary structure of raw chickpea proteins (CPI) are like those reported by other authors [37]. The elderly salivary conditions significantly modified (*p* < 0.05) the α-helix secondary structure of both protein sources; thus, the α-helix structure of the native proteins decreased by ~36% in CPI and ~29% in WPI, while total β-sheet structure increased by ~36% and ~85%, and β-turn increased by ~16% and decreased 30% in CPI and WPI, respectively. The highest percentage of β-sheet was found in both protein isolates treated by elderly salivary conditions, CPI-EL and WPI-EL. Likewise, CPI exhibited higher percentages of β-turn than WPI in all samples studied. Interestingly, the unordered random coil state increased slightly after oral processing (CPI-A < CPI-EL). Even so, this increase was significantly higher (*p* < 0.05) in whey proteins treated under elderly oral conditions (WPI-A < WPI-EL) than in adults and their chickpea counterparts, confirming the results determined by SDS-PAGE and CD analysis. These structural results suggest that the α-helix structures were converted to β-turns in CPI and to a random coil state in WPI during oral processing. Accordingly, high contents of rigid secondary structure fractions, such as β-sheet and β-turns, have been associated with lower in vitro protein digestibility [40].

In addition, the effect of the food matrix was lost owing to the alkaline extraction of the protein isolates, which included defatting. Prodic et. al. (2018) [41] established that pure proteins can have different sensitivities to proteolysis at the gastric level compared to proteins adsorbed at the oil-in-water interface of the food matrix owing to changes in the protein structure. Thus, at the oral level, sensitivity and differences can also be presented. Based on the protein characteristics obtained by PAGE, CD, and Raman spectroscopy analyses, it was hypothesized that a higher potassium chloride content, together with a higher amount of α-amylase in the simulated saliva of the elderly and that α-amylase may contain impurities, affects the conformational stability of whey proteins at the oral level more than chickpea proteins. It is presumed that a scattered distribution of ion density around the α-helix in whey proteins could occur during oral processing, and that high ionic strength promotes further disruption of the α-helix, which has a more open structure than the β-sheet.

## 4. Conclusions

Changes in simulated saliva of the elderly related to alkaline pH, high electrolyte concentration, and increased α-amylase activity influence the surface and conformational properties of chickpea and whey proteins. The negative surface charge decreased for both proteins, and orally treated proteins with lower molecular weights were identified. The chickpea proteins shifted towards larger protein particle distributions, while their water absorption index decreased.

High ionic strength in the simulated saliva of the elderly impacted the ordered structure of both sources of proteins, losing the structural features and decreasing α-helix for chickpea and whey proteins, and a significant increase in the unordered random coil state was observed. The results demonstrated that the structural characteristics of chickpea proteins were more stable than those of whey proteins due to their higher molecular weight, more rigid structure, less charged surface, and different amino acid composition. It has been hypothesized that high ionic strength may affect the ion density around the α-helix structure and may destabilize it. A decrease in α-helix and an increase in rigid structures such as β-sheet and β-turn at the oral level could subsequently lead to lower chickpea protein digestibility at the gastrointestinal level in the elderly population. Our findings could contribute to pinpointing the differences in protein structural changes between plant and milk sources. The latter is important because plant-based proteins are well-established alternatives to animal proteins. In addition, it can help to improve the understanding of the oral processing for adults and the elderly and provides new insight of value for future recommendations for elderly individuals.

## Figures and Tables

**Figure 1 foods-12-03668-f001:**
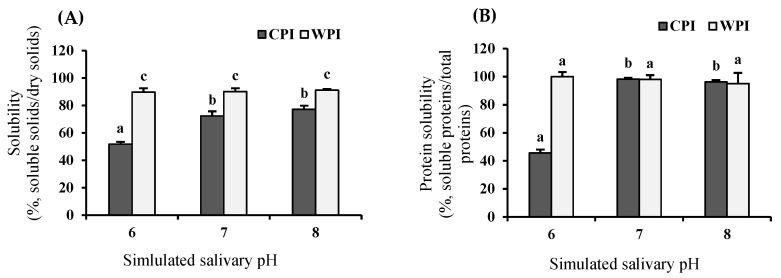
(**A**) Solubility (%, g soluble solids/g total solids ×100) and (**B**) Protein solubility (%, g soluble proteins/g total proteins ×100) for CPI and WPI at three pH of simulated salivary fluid (6.0, 7.0, and 8.0) at 37 °C. Data are shown as a mean ± SD (*n* = 3). Different letters on each bar indicate significant differences (*p* < 0.05) between pH values for each type of protein.

**Figure 2 foods-12-03668-f002:**
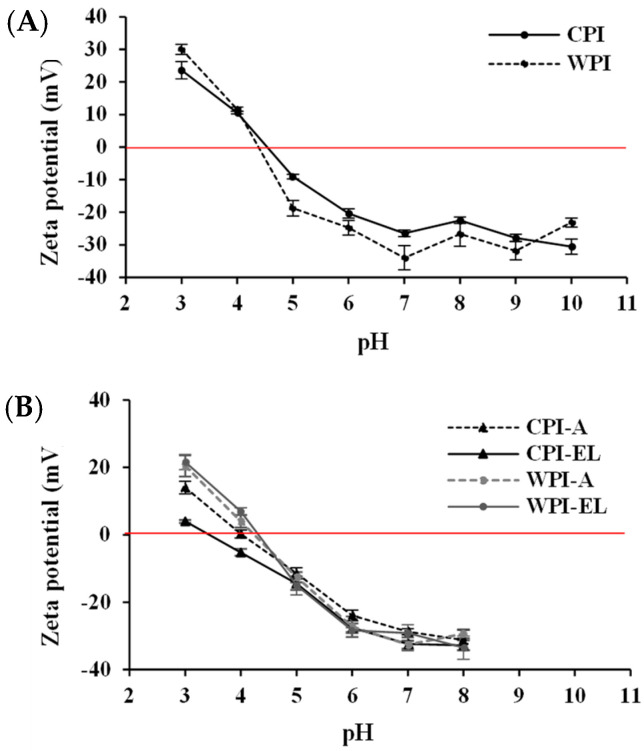
Zeta potential (mV)-pH profiles of (**A**) raw CPI and WPI, and (**B**) oral-treated CPI and WPI (-A = Adult and -EL = Elderly simulated salivary conditions) at 37 °C. Data are shown as the mean ± SD (*n* = 9).

**Figure 3 foods-12-03668-f003:**
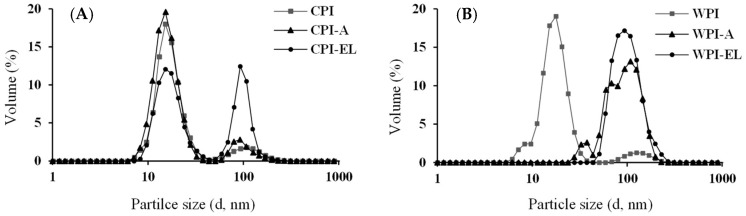
Particle size distribution (d, nm) of protein isolates from (**A**) CPI and (**B**) WPI before and after in vitro oral processing (-A: Adult and -EL: Elderly oral conditions) measured at 37 °C. Data represent the mean + SD (*n* = 30).

**Figure 4 foods-12-03668-f004:**
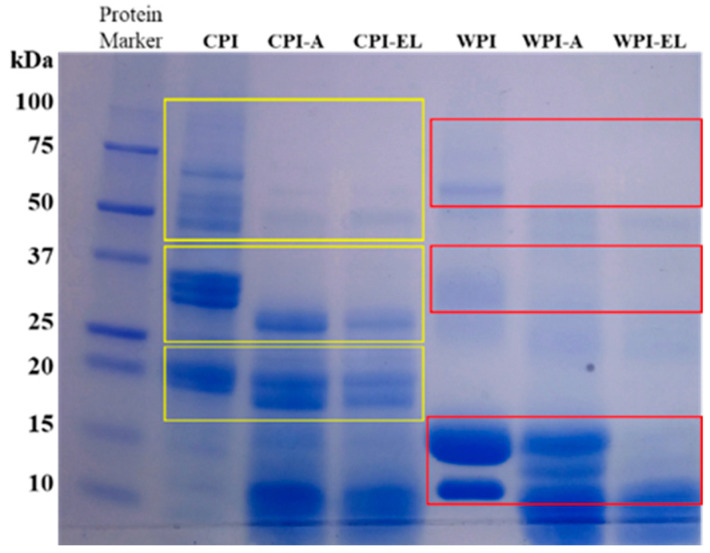
SDS-PAGE profiles of CPI (line 2, 3, 4) and WPI (line 5, 6, 7) before and after in vitro oral processing (-A: Adult and -EL: Elderly oral conditions). The numbers on the left represent the molecular weights of protein markers (kDa, line 1).

**Figure 5 foods-12-03668-f005:**
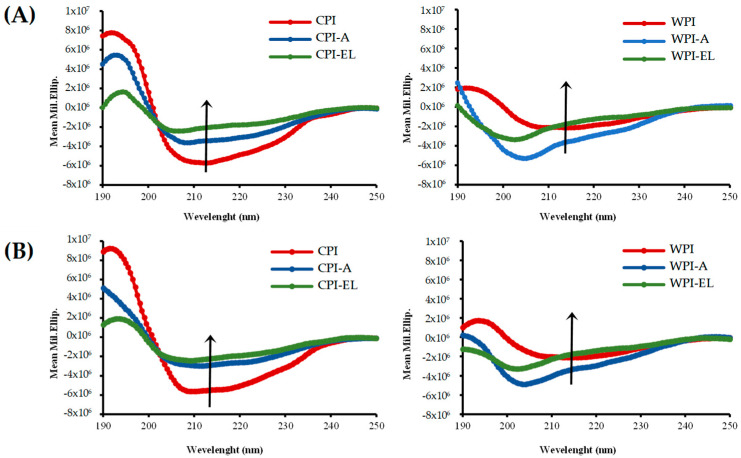
Representative CD spectra of proteins from CPI and WPI before and after oral in vitro processing (-A: Adult and -EL: Elderly oral conditions) measured at 190–250 nm and run at (**A**) 25 °C and (**B**) 37 °C.

**Table 1 foods-12-03668-t001:** Percentage of the secondary structure of the chickpea proteins (CPI) and whey proteins (WPI) before and after in vitro oral digestion simulating adult (-A) and elderly (-EL) oral conditions.

Secondary Structure (%)	CPI	CPI-A	CPI-EL	WPI	WPI-A	WPI-EL
α-helix	44.5 ± 5 ^b^	42.2 ± 9 ^b^	28.5 ± 5 ^a^	40.2 ± 2 ^b^	31.5 ± 4 ^a^	28.5 ± 3 ^a^
Total β-sheet	26.0 ± 5 ^ab^	28.2 ± 9 ^abc^	35.3 ± 5 ^cd^	21.3 ± 3 ^a^	31.7 ± 6 ^bcd^	39.3 ± 6 ^d^
β-turn	28.6 ± 2 ^bc^	31.7 ± 7 ^bc^	33.1 ± 9 ^c^	26.2 ± 5 ^abc^	25.0 ± 7 ^ab^	18.3 ± 4 ^a^
Random coil	4.5 ± 2 ^a^	3.4 ± 2 ^a^	9.8 ± 1 ^b^	3.2 ± 2 ^a^	11.2 ± 3 ^b^	15.2 ± 4 ^c^

Results (mean of *n* = 6) were obtained from Raman spectra after deconvolution of the amide I band. Different letters in the same row indicate significant differences (*p* < 0.05).

## Data Availability

Data is contained within the article.

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
