# Peer review of "Conformational and Structural Changes in Chickpea Proteins Caused by Simulated Salivary Alterations in the Elderly"

_foods, 2023, doi:10.3390/foods12193668_

Round 1

Reviewer 1 Report

The publication contains important information from the point of view of food technology on conformational and structural changes in chickpea proteins depending on pH. This publication seems to be within the scope of journal. However, it seems that the authors had not attached the final version of publication but earlier version. For this reason, it needs several corrections.

1.     Line 64: it should be „pI (pH ~5.0)” instead of „pI (~pH 5.0)”.

2.     Please correct incorrect chemical symbols in the whole manuscript (e.g. lines 75, 418). It should be “Cl-“ and “Ca2+” instead of “Cl-“ and “Ca++”.

3.     Line 110: It should be „mL” instead of „ml”.

4.     Line 119: Hexane was mixed with what in a 5:1 ratio? The text should include information whether it is a proportion by weight or volume?

5.     Line 122 and others: Bibliographic data regarding AOAC (2019) [18] should enable access to these materials, especially since this reference is crucial for the methodology of manuscript.

6.     Line 146: Please add information about eluents used during amino acids analysis by HPLC. If analysis was not isocratic, please add information about program.

7.     Lines 201, 204: It should be „cm-1 instead of „cm-1”.

8.     Line 290: The authors wrote that the decrease in the surface charge of CPI may be influenced by other components (e.g. lipids) and referred to table S2, but this table does not contain information about the composition of lipids. Please add appropriate information.

Author Response

Response to reviewer 1 comments

Thank you very much for taking the time to review this manuscript. Please find the detailed responses below, and the corresponding revisions and corrections highlighted in the re-submitted files (manuscript).

Questions for General Evaluation

Reviewer’s Evaluation

Response and Revisions

Does the introduction provide sufficient background and include all relevant references?

Yes

Are all the cited references relevant to the research?

Must be improved

Thank you for pointing this out. Therefore, a new reference has been introduced in the manuscript, highlighting the changes in yellow

Is the research design appropriate?

Yes

Are the methods adequately described?

Must be improved

Thank you for pointing this out. We agree with these comments. Therefore, all suggestions for improvements and changes have been introduced in the manuscript, highlighting the changes in yellow

Are the results clearly presented?

Yes

Are the conclusions supported by the results?

Yes

Responses

  1. Line 64: it should be „pI (pH ~5.0)” instead of „pI (~pH 5.0)”. Response: The text has been changed as suggested in Line 66
  1. Please correct incorrect chemical symbols in the whole manuscript (e.g. lines 75, 418). It should be “Cl-“ and “Ca2+” instead of “Cl-“ and “Ca++”. Response: The text has been changed as suggested in Lines 78, 353, 438
  1. Line 110: It should be „mL” instead of „ml”. Response: The text has been changed as suggested in Line 114
  1. Line 119: Hexane was mixed with what in a 5:1 ratio?The text should include information whether it is a proportion by weight or volume? Response: The text has been modified as suggested making clearer regarding the solutions proportions. The text has been changed in line 123
  1. Line 122 and others: Bibliographic data regarding AOAC (2019) [18] should enable access to these materials, especially since this reference is crucial for the methodology of manuscript.

Response

We appreciate the reviewer's suggestions. We agree with these comments. The relevant details of the proximal analysis methodologies using AOAC as a reference are described in the methodology section (Section 2.3), as well as the time and temperature parameters for the analysis of moisture content.

Please refer to lines 137-143

  1. Line 146: Please add information about eluents used during amino acids analysis by HPLC. If analysis was not isocratic, please add information about program.

Response:

Thank you for pointing this out. We agree with these comments. Therefore, the required details have been described in the methodology (Section 2.4).

Please refer to lines 150-154.

  1. Lines 201, 204: It should be „cm-1 instead of „cm-1”.

Response:

We appreciate the reviewer's suggestions for the improvement of the manuscript. This suggestion has been introduced in the manuscript, highlighting the changes in the yellow color.

Please refer to lines 214, 217

  1. Line 290: The authors wrote that the decrease in the surface charge of CPI may be influenced by other components (e.g. lipids) and referred to table S2, but this table does not contain information about the composition of lipids. Please add appropriate information.

Response:

Thank you for pointing this out. We appreciate the reviewer's suggestions for the improvement. The lipid contents of CPI and WPI are given in Table S2 as “Fat,”. Therefore, in Table S2, this labeling (“Fat”) has been modified to “Lipid” to avoid confusion.

Please refer to line 703

Reviewer 2 Report

General comment:

The work of Ingrid Contardo*, Fanny Guzmán and Javier Enrione, entitled: ”Conformational and structural changes in chickpea proteins caused by simulated salivary alterations in the elderly”, aimed to characterize differences in INFOGEST2.0-simulated oral digestion protocol in elderly and normal, adult (control) populations with chickpeas protein solutions (CPI) obtained from n-hexane defatted chickpea flour of the specific particle range. They have used SDS-PAGE to observe protein profiles of defatted protein solution of CPI and whey, Raman and CD spectroscopy to search for differences in secondary structures and other associated physical and biochemical properties among oral protein digesta.

The article is interesting, as data on sole oral phase of simulated digestion are very scarce, due to the fact that most researchers conduct consecutive steps and assess most frequently the end-digestion products related to oral-gastric-intestinal phase, less data are generated after the oral-gastric phase, and the least data for oral phase that lasts 2 minutes. Yet the oral phase can have profound effects per se, on protein profiles, even though nature did not design it to tackle with enzymatic protein hydrolysis. The reasons are well elaborated in the review paper:” Antioxidant Properties of Protein-Rich Plant Foods in Gastrointestinal Digestion—Peanuts as Our Antioxidant Friend or Foe in Allergies” published in Antioxidants in 2023.

However certain specific major and minor comments should be attended and concerns clarified and better described and explained.

1. Major comment all about methodology and missing controls

Please describe explicitly conditions of SDS-PAGE experiment. How 38 ug of protein was measured and why authors have adopted the equal amount approach? Usually, samples are loaded in respect to equal volumes (which is important and correct approach if one wants to examine differences in digestion profiles, since the starting material quantity is the same and applied final volumes of SSF are the same).

In addition, when observing the Figure 4, it cannot be that quantities of proteins were applied all as 38 ug since there is much less in lanes 4 and 7 in respect to their adult population. The methodological approach and the entire reasoning here is at least unclear if not incorrect. I will shortly explain my reasoning:

1) If authors based their protein load calculations on BCA or Bradford measurements then they should show the entire SDS-PAGE with CBB dye front and below to see if there is accumulation of di, tri and tetra peptides that are smaller in MW then dye and were running below the front, which would explain huge differences in protein amounts between 3&4 and 6&7 lane pairs.

2) The most exact approach would the equal volume application of oral digesta phases. This is mandatory and the equal amounts e.g. 38 ug of each sample, this is something the protein biochemist would call “an option to look for relative differences”. However if the first was not done, we cannot know the true scope of the effects of simulated oral digestion on shares of specific protein identities and their true profiling. As later authors were stated that upon protein composition and amount all their variables under study are dependent.

2. There is no control sample of digestion established. I am speaking about the control that is comprised of everything else except of replacing alfa amylase with the equal volume of SSF. So instead of having 4 samples here (CPI-A, CPI-E, whey-A, whey-E), there should be 8 of them since each of them should have its control. Why is that so? Perhaps the best proof is reflected in our current research, with piece of information shown below.

The first lane are protein mass markers, second lane is the control of the digestion, and the third lane is true digestion sample. Lanes 2 and 3 are reflecting the oral phase of raw peanut, and it full food matrix.

3. Authors should clarify why they have opted to work with defatted chickpea flour? Is it common commercial practice to remove lipids from the chickpea flour (defatted flour) prior to its use in food industry? If so, then this should be mentioned and documented/supported with the relevant references. If not, then, authors should explain and rebut the reason for conducting the research in this manner that is not immanent to the directions of INFOGEST protocol, especially not its variant 2.0 that authors are calling upon. To learn about compliance to INFOGEST protocol and the effects of defatting on the upstream and downstream processing of the oral/gastric and gastro intestinal digesta please consider reading more references

4. The effect of true and realistic food matrix has been lost due to defatting. Please mention relevant limitations of your study when explaining your results. To learn about this effect from the similar food source and within the original research article, please read the fourth paragraph of the discussion of “Influence of peanut matrix on stability of allergens in gastric-simulated digesta: 2S albumins are main contributors to the IgE-reactivity of short digestion resistant peptides.

Figure 1

1. Minor comments

Abstract, lines 20-21 “Under simulated salivary elderly conditions, the molecular ordered structure was significantly affected, decreasing α-helix by ~36% and ~29% for CPI and WPI, respectively”.

This sentence should be changed into something like: ”Under condition of simulated oral digestion in elderly population, the ordered secondary protein structure was significantly affected, decreasing α-helix by ~36% and ~29% in CPI and WPI compared to control (adult) population, respectively.”

Regarding the English language, aside of moderate level needed to repair style and grammar, I have one specific request for an expression to be changed throughout the manuscript. Instead of using the expression of “simulated salivary pH” (there are 6 cases in the manuscript), replace them with “pH of simulated salivary fluid” or “pH of SSF”.  This is because authors were dealing with the real pH not the simulated pH, while only saliva was simulated or in other words simulated salivary fluid-SSF”.

Regarding the English language, aside of moderate level needed to repair style and grammar, I have one specific request for an expression to be changed throughout the manuscript. Instead of using the expression of “simulated salivary pH” (there are 6 cases in the manuscript), replace them with “pH of simulated salivary fluid” or “pH of SSF”.  This is because authors were dealing with the real pH not the simulated pH, while only saliva was simulated or in other words simulated salivary fluid-SSF”. 

Author Response

Response to reviewer 2 comments

Thank you very much for taking the time to review this manuscript. Please find the detailed responses below, and the corresponding revisions and corrections highlighted in the re-submitted files.

Questions for General Evaluation

Reviewer’s Evaluation

Response and Revisions

Does the introduction provide sufficient background and include all relevant references?

Can be improved

Thank you for pointing this out.

Are all the cited references relevant to the research?

Yes

Is the research design appropriate?

Can be improved

Thank you for pointing this out. We agree with these comments. Therefore, all suggestions for improvements and changes have been introduced in the manuscript, highlighting the changes in yellow

Are the methods adequately described?

Must be improved

We agree with these comments. Therefore, all suggestions for improvements and changes have been introduced in the manuscript, highlighting the changes in yellow

Are the results clearly presented?

Can be improved

Are the conclusions supported by the results?

Yes

  1. Major comment all about methodology and missing controls
  • Please describe explicitly conditions of SDS-PAGE experiment. How 38 ug of protein was measured and why authors have adopted the equal amount approach? Usually, samples are loaded in respect to equal volumes (which is important and correct approach if one wants to examine differences in digestion profiles, since the starting material quantity is the same and applied final volumes of SSF are the same). In addition, when observing the Figure 4, it cannot be that quantities of proteins were applied all as 38 ug since there is much less in lanes 4 and 7 in respect to their adult population. The methodological approach and the entire reasoning here is at least unclear if not incorrect. I will shortly explain my reasoning:

If authors based their protein load calculations on BCA or Bradford measurements then they should show the entire SDS-PAGE with CBB dye front and below to see if there is accumulation of di, tri and tetra peptides that are smaller in MW then dye and were running below the front, which would explain huge differences in protein amounts between 3&4 and 6&7 lane pairs.

The most exact approach would the equal volume application of oral digesta phases. This is mandatory and the equal amounts e.g. 38 ug of each sample, this is something the protein biochemist would call “an option to look for relative differences”. However if the first was not done, we cannot know the true scope of the effects of simulated oral digestion on shares of specific protein identities and their true profiling. As later authors were stated that upon protein composition and amount all their variables under study are dependent.

Response:

Thank you for pointing this out. We appreciate the reviewer's suggestions in order to improve and clarify the methodology used in this study. The reported 38 µg of proteins corresponded to 38 µg of protein in suspension of sample diluted with sample buffer and 2-mercaptoethanol.

The samples were diluted with distilled water to a final concentration of 3 mg protein/mL. The samples, with the same volume (30 µL), were produced by mixing them 1:1 with sample buffer and 10 µL 2-mercaptoethanol. All samples were heated at 100°C for 5 min before loading. Subsequently, 10 μL of the sample solution was loaded into an electrophoresis lane. Standard molecular weight markers within the 10–250 kDa range were used (KaleidoscopeTM, Precision Plus Protein StandardsTM; Bio-Rad). Electrophoresis was run at 90V, and the resulting gel was stained with 0.25% Coomassie blue G250. This clarification was incorporated into the methodology (Section 2.7.1.). Please refer to lines 189-193

The final concentration of 3 mg protein/mL and the volume of sample (10 µL) used to run the SDS-PAGE experiment was based on literature references [1, 2, 3] and Bio-Rad recommendations (in the Protein Electrophoresis Guide).

References.

  1. [1] Sanchón, J., Fernández-Tomé, S., Miralles, B., Hernández-Ledesma, B., Tomé, D., Gaudichon, C., & Recio, I. (2018). Protein degradation and peptide release from milk proteins in human jejunum. Comparison with in vitro gastrointestinal simulation. Food Chemistry, 239, 486-494.
  2. [2] del Rio, A. R., Opazo-Navarrete, M., Cepero-Betancourt, Y., Tabilo-Munizaga, G., Boom, R. M., & Janssen, A. E. (2020). Heat-induced changes in microstructure of spray-dried plant protein isolates and its implications on in vitro gastric digestion. Lwt, 118, 108795.
  3. [3] Bilawal, A., Gantumur, M. A., Huang, Y., Qayum, A., Ishfaq, M., & Jiang, Z. (2023). Effect of transglutaminase cross-linking and emulsification on the stability and digestion of limonene emulsions. Process Biochemistry, 131, 210-216.
  • There is no control sample of digestion established. I am speaking about the control that is comprised of everything else except of replacing alfa amylase with the equal volume of SSF. So instead of having 4 samples here (CPI-A, CPI-E, whey-A, whey-E), there should be 8 of them since each of them should have its control. Why is that so? Perhaps the best proof is reflected in our current research, with piece of information shown below. The first lane are protein mass markers, second lane is the control of the digestion, and the third lane is true digestion sample. Lanes 2 and 3 are reflecting the oral phase of raw peanut, and it full food matrix.

Response:

Thank you for pointing this out. We agree with these comments. In the study, we did not use a control that comprised each case/source since we wanted to test, except for replacing α-amylase with an equal volume of SSF. Whey protein isolate was used as a reference because we compared the differences between two different sources: CPI and WPI. In addition, we compared two oral conditions that showed differences in protein structure of both protein sources linked to variations in environmental alterations in terms of pH, electrolyte concentration, and alpha amylase activity, using “adult” oral conditions as control. Thus, we demonstrated that a higher potassium chloride content, together with a higher amount of α-amylase in the simulated saliva of the elderly, and that α-amylase may contain impurities, affect the conformational stability of whey proteins at the oral level more than chickpea proteins.

  • Authors should clarify why they have opted to work with defatted chickpea flour? Is it common commercial practice to remove lipids from the chickpea flour (defatted flour) prior to its use in food industry? If so, then this should be mentioned and documented/supported with the relevant references. If not, then, authors should explain and rebut the reason for conducting the research in this manner that is not immanent to the directions of INFOGEST protocol, especially not its variant 2.0 that authors are calling upon. To learn about compliance to INFOGEST protocol and the effects of defatting on the upstream and downstream processing of the oral/gastric and gastrointestinal digesta please consider reading more references.

Response:

Thank you for pointing this out. We appreciate the reviewer's suggestions for improving and clarifying the methodologies used in this study. However, in our study, the analyzed sample corresponded to a chickpea protein isolate (CPI) and not chickpea flour. Additionally, a commercial whey protein isolate (WPI) was used as reference. The extraction process of CPI corresponds to the alkaline extraction of proteins, which involves a lipid extraction process, followed by precipitation through the isoelectric point (as explained in Section 2.2) and freeze-drying process. We started the protein extraction process from peeled commercial chickpeas and applied grinding to achieve greater control of the particle size used. The fraction of flour enriched in proteins (212–300 µm) was defatted and used (this was mentioned in the line 121). Subsequently, a suspension of these freeze-dried protein isolates was prepared at 10% (w/v), as described in the methodology (Section 2.8). Therefore, it is not part of the study to understand the effect of the matrix in which chickpea proteins were found (grain), as the food matrix in this case was an aqueous suspension (as a vegetable drink would be), although the consensus paper guided by Brodkorb et al. (2019), undertaken during the COST INFOGEST action, suggests the use of the whole food in the digestion assays to obtain more realistic results of what may occur in the gastrointestinal system. However, this study aimed to understand how protein isolates in solutions can be affected by changes in simulated saliva of the elderly. Several studies have analyzed the digestion of protein isolates using the Infogest protocol, in which legume flours are defatted to extract proteins.

Some references

  1. Brodkorb, A., Egger, L., Alminger, M., Alvito, P., Assunção, R., Ballance, S., ... & Recio, I. (2019). INFOGEST static in vitro simulation of gastrointestinal food digestion. Nature protocols, 14(4), 991-1014.
  2. Santos-Hernández, M., Alfieri, F., Gallo, V., Miralles, B., Masi, P., Romano, A., ... & Recio, I. (2020). Compared digestibility of plant protein isolates by using the INFOGEST digestion protocol. Food Research International, 137, 109708.
  3. Fu, L., Gao, S., & Li, B. (2023). Impact of Processing Methods on the In Vitro Protein Digestibility and DIAAS of Various Foods Produced by Millet, Highland Barley and Buckwheat. Foods, 12(8), 1714.
  4. Sousa, R., Portmann, R., Dubois, S., Recio, I., & Egger, L. (2020). Protein digestion of different protein sources using the INFOGEST static digestion model. Food Research International, 130, 108996.
  • The effect of true and realistic food matrix has been lost due to defatting. Please mention relevant limitations of your study when explaining your results. To learn about this effect from the similar food source and within the original research article, please read the fourth paragraph of the discussion of “Influence of peanut matrix on stability of allergens in gastric-simulated digesta: 2S albumins are main contributors to the IgE-reactivity of short digestion resistant peptides. Figure 1

Response:

Thank you for pointing this out. We appreciate the reviewer's suggestions. Therefore, we have included a comment on pure proteins in solution that can have different sensitivities to proteolysis compared to proteins adsorbed at the oil-in-water interface owing to changes in the protein structure. Precisely, the latter is the focus of this study, to characterize how the structure of pure proteins in solution changes in oral processing.

Please refer to lines 476 – 480

  1. Minor comments
  • Abstract, lines 20-21 “Under simulated salivary elderly conditions, the molecular ordered structure was significantly affected, decreasing α-helix by ~36% and ~29% for CPI and WPI, respectively”. This sentence should be changed into something like: ”Under condition of simulated oral digestion in elderly population, the ordered secondary protein structure was significantly affected, decreasing α-helix by ~36% and ~29% in CPI and WPI compared to control (adult) population, respectively.”

Response:

We appreciate the reviewer's suggestions for the improvement of the manuscript. Therefore, the suggested changes have been incorporated into the abstract highlighted in yellow.

Please refer to lines 21-23.

  • Regarding the English language, aside of moderate level needed to repair style and grammar, I have one specific request for an expression to be changed throughout the manuscript. Instead of using the expression of “simulated salivary pH” (there are 6 cases in the manuscript), replace them with “pH of simulated salivary fluid” or “pH of SSF”.  This is because authors were dealing with the real pH not the simulated pH, while only saliva was simulated or in other words simulated salivary fluid-SSF”.

Response:

Thank you for pointing this out. We agree with these comments. Therefore, the suggested changes have been incorporated into the manuscript highlighted in yellow.

Please refer to lines to the following lines:

Lines 239

Line 247

Lines 252-253

Line 277

Line 335

Lines 343-344

Line 517

Line 695

Reviewer 3 Report

The title is not representative of the content, since also whey protein are analyzed

Abstract and following:

“plant-based protein”

plant protein.

line 33

“they lack some essential amino acids”

it’s incorrect: they contain lower amount of some limiting aminoacid

Legumes are limited only in sulfur aminoacid

Abstract: it should be improved in its clarity

Introduction

should be more concise

Material and methods are well detailed

Results should be exposed separately from discussion: results should be detailed, while in the discussion summarily recalled in order to comment them

Conclusion repeat some results, which is redundant

the article requires major revisions

Author Response

Response to reviewer 3 comments

Thank you very much for taking the time to review this manuscript of the manuscript. Please find the detailed responses below, and the corresponding revisions and corrections highlighted in the re-submitted files. All suggestions for improvements and changes have been introduced in the manuscript, highlighting the changes in yellow.

Questions for General Evaluation

Reviewer’s Evaluation

Response and Revisions

Does the introduction provide sufficient background and include all relevant references?

Can be improved

We agree with these comments. Accordingly, we have incorporated some modifications to the manuscript, highlighting the changes in yellow.

Are all the cited references relevant to the research?

Yes

Is the research design appropriate?

Can be improved

We agree with these comments. Accordingly, we have incorporated some modifications to the manuscript, highlighting the changes in yellow.

Are the methods adequately described?

Yes

Are the results clearly presented?

Can be improved

We agree with these comments. Accordingly, we have incorporated some modifications to the manuscript, highlighting the changes in yellow.

Are the conclusions supported by the results?

Yes

Comments

  • The title is not representative of the content, since also whey protein are analyzed

Response:

We appreciate the reviewer's suggestions for the improvement of the manuscript. Thanks for pointing this out. Accordingly, we have revised the title. However, we wanted to retain this initial title: “Conformational and structural changes in chickpea proteins caused by simulated salivary alterations in the elderly”, because we wanted to highlight studies on chickpea proteins, as there is little information about their processing at the oral level in simulated elderly conditions. In addition, whey proteins have only been used as reference proteins because they have been widely studied and analyzed at the gastrointestinal level.

  • Abstract and following:

“plant-based protein” plant protein.

 Response:

We appreciate the reviewer's suggestions for the improvement of the manuscript. We agree with these comments. Accordingly, we have modified the manuscript, highlighting the changes in yellow.

Please refer to line 508

  • line 33 “they lack some essential amino acids” it’s incorrect: they contain lower amount of some limiting aminoacid. Legumes are limited only in sulfur aminoacid

Response:

Thank you for pointing this out. We agree with these comments. Accordingly, we have incorporated some modifications to the manuscript, highlighting the changes in yellow. Please refer to line 34

  • Abstract: it should be improved in its clarity

Response:

We appreciate the reviewer's suggestions for the improvement of the manuscript. Accordingly, we have improved the abstract, highlighting the changes in yellow.

Please refer to lines 21-23

  • Introduction should be more concise

Response:

We appreciate the reviewer's suggestions for the improvement of the manuscript. Thank you for pointing this out. The proposed introduction focuses on the precise details of oral changes in terms of electrolytes and pH. In addition, details have been mentioned about conformational changes in proteins are affected by alterations in the electrolytes and pH.

  • Material and methods are well detailed

Response: Thank you for pointing this out.

  • Results should be exposed separately from discussion: results should be detailed, while in the discussion summarily recalled in order to comment them

Response:

Thank you for pointing this out. We believe that since the results were presented in an integrated manner with the discussions, it allows for a better understanding and visualization of the results produced, because the study involves several variables and responses. In addition, it permits better integration also between the results and figures.   

  • Conclusion repeat some results, which is redundant

Response:

We appreciate the reviewer's suggestions for the improvement of the manuscript. We agree with these comments. The α-helix percentages were eliminated. Please refer to lines 497-498. The repetition of some general results reinforces the idea that changes at the oral level could negatively affect subsequent gastrointestinal digestion.

  • the article requires major revisions

Response:

Thank you for pointing this out to us. Thank you for taking the time reviewing the manuscript. Please find the revisions/corrections 

Round 2

Reviewer 2 Report

Dea all

thank you for accepting majority of my comments, and clarofying the rest. I have no furhter questions.

Kind regards.

The English language grammar is acceptable.

Reviewer 3 Report

I see that the Authors have answered the questions posed and I have no objection to the revised version of the manuscript. which can be accepted it in its current version for publication in the journal.